Analysis of dynamic and widespread lncRNA and miRNA expression in fetal sheep skeletal muscle

Yuan Chao 1
Zhang Ke 2
Yue Yaojing 1
Guo Tingting 1
Liu Jianbin 1
Niu Chune 1
Sun Xiaoping 1
Feng Ruilin 1
Wang Xiaolong 2
Yang Bohui yangbohui@caas.cn 1
1 Sheep Breeding Engineering Technology Research Center, Lanzhou Institute of Husbandry and Pharmaceutical Sciences of Chinese Academy of Agricultural Sciences , Lanzhou , Gansu , China
2 College of Animal Science and Technology, Northwest A&F University , Yangling , Shaanxi , China
Posner Mason
Electronic publication date: 2020 Sep 22
Publication date: 2020
Volume: 8
Electronic Location ID: e9957
Received 2019 Nov 22; Accepted 2020 Aug 25
Copyright: ©2020 Yuan et al.
Copyright year: 2020
Copyright holder: Yuan et al.
License: This is an open access article distributed under the terms of the Creative Commons Attribution License, which permits unrestricted use, distribution, reproduction and adaptation in any medium and for any purpose provided that it is properly attributed. For attribution, the original author(s), title, publication source (PeerJ) and either DOI or URL of the article must be cited.
License URL: https://creativecommons.org/licenses/by/4.0/

Keywords: lncRNA, miRNA, Transcriptome, Longissimus dorsi muscle, Fetal sheep

Funding: Agricultural Science and Technology Innovation Program of China CAAS-ASTIP-2015-LIHPS the Selection of Scientific Research Topics for Significant Production of the Chinese Academy of Agricultural Sciences CAAS-ZDXT2018006 the Modern China Wool Cashmere Technology Research System CARS-39-02 Central Level, Scientific Research Institutes for Basic R&D Special Fund Business 1610322019011 This work was supported by the Agricultural Science and Technology Innovation Program of China (CAAS-ASTIP-2015-LIHPS), the Selection of Scientific Research Topics for Significant Production of the Chinese Academy of Agricultural Sciences (CAAS-ZDXT2018006), the Modern China Wool Cashmere Technology Research System (CARS-39-02), Key Research the Development Program of Gansu Provincial-Agricultural Project (17YF1NA069), and the Central Level, Scientific Research Institutes for Basic R&D Special Fund Business (1610322019011). The funders had no role in study design, data collection and analysis, decision to publish, or preparation of the manuscript.

==============================
The sheep is an economically important animal, and there is currently a major focus on improving its meat quality through breeding. There are variations in the growth regulation mechanisms of different sheep breeds, making fundamental research on skeletal muscle growth essential in understanding the regulation of (thus far) unknown genes. Skeletal muscle development is a complex biological process regulated by numerous genes and non-coding RNAs, including microRNAs (miRNAs) and long non-coding RNAs (lncRNAs). In this study, we used deep sequencing data from sheep longissimus dorsi (LD) muscles sampled at day 60, 90, and 120 of gestation, as well as at day 0 and 360 following birth, to identify and examine the lncRNA and miRNA temporal expression profiles that regulate sheep skeletal myogenesis. We stained LD muscles using histological sections to analyse the area and circumference of muscle fibers from the embryonic to postnatal development stages. Our results showed that embryonic skeletal muscle growth can be characterized by time. We obtained a total of 694 different lncRNAs and compared the differential expression between the E60 vs. E90, E90 vs. E120, E120 vs. D0, and D0 vs. D360 lncRNA and gene samples. Of the total 701 known sheep miRNAs we detected, the following showed a wide range of expression during the embryonic stage: miR-2387, miR-105, miR-767, miR-432, and miR-433. We propose that the detected lncRNA expression was time-specific during the gestational and postnatal stages. GO and KEGG analyses of the genes targeted by different miRNAs and lncRNAs revealed that these significantly enriched processes and pathways were consistent with skeletal muscle development over time across all sampled stages. We found four visual lncRNA–gene regulatory networks that can be used to explore the function of lncRNAs in sheep and may be valuable in helping improve muscle growth. This study also describes the function of several lncRNAs that interact with miRNAs to regulate myogenic differentiation.

Introduction

The first three stages of mammalian muscle development are completed during the embryonic stage, and the number of muscle fibers generally does not change after birth. Postnatal muscle growth is mainly triggered by muscle fiber hypertrophy and increased intermuscular fat (Thomas & Mathias, 2011). In modern animal husbandry, skeletal muscle is considered the most economically important part of an animal’s body. Skeletal muscle yield and quality are determined by the animal’s muscle fiber type, metabolism, and physiological characteristics. Recent research has been key to understanding the molecular mechanism of skeletal muscle formation, and the important roles that myogenic regulatory factors (MRFs) and myocyte enhancer factor 2 (MEF2) protein families play in skeletal muscle development (Lang et al., 2007; Snyder et al., 2013).

It is now recognized that the myogenic process involves more than just the protein-coding gene signaling pathways. microRNAs (miRNAs) are the most widely studied class of non-coding RNA molecules that participate in muscle production (Wen, Nie & Zhang, 2013), but several recent studies have shown that long non-coding RNAs (lncRNAs) may also play a role in muscle differentiation (Gong et al., 2015; Li et al., 2017). LncRNAs are widely found in mammals; have time-, space-, and tissue-specific expression; and have been shown to contribute to multiple processes, including epigenetic, transcriptional, and post-transcriptional regulation (Ming et al., 2016). lncRNA-tiny non-coding RNAs (TncRNAs) found in the porcine fetal trophoblast have been found to be upregulated in embryonic skeletal muscle, and in Tongcheng and Changbai pigs, researchers observed expression level differences in fetal skeletal muscle on the 90th day of pregnancy, suggesting that lncRNAs may have an effect on the embryonic development of porcine skeletal muscle (Ren et al., 2010). Transcriptome sequencing analysis of bovine longissimus dorsi (LD), scapula, intercostal, and gluteal muscles revealed that in bovine myoblasts, the lncRNA lncYYW positively regulated the expression of the growth hormone 1 (GH1) gene and its downstream genes AKT1 and PIK3CD. lncYYW was upregulated during myoblast differentiation, and its overexpression increased the number of cells during the S phase of the cell cycle (Yue et al., 2017). Transcriptome sequencing of 45-, 60-, and 105-day-old goat fetuses and 3-day-old lamb LD muscle tissue identified 3,981 lncRNAs that were highly conserved in all four stages. A two-time-point comparison found 577 differentially expressed lncRNAs that may have specific biological effects on early goat muscle development (Zhan et al., 2016). A previous study used strand-specific Ribo-Zero RNA technology to sequence the LD muscles of Hu sheep at three important developmental stages (fetus, lamb, and adult), and obtained a total of 6,924 lncRNAs. The differentially expressed lncRNAs were shown to contribute to biological processes during the embryonic stage, including organ morphogenesis and skeletal and muscle development. This study was the first to systematically analyze lncRNAs in Hu sheep muscles and deliver valuable information on sheep muscle development (Shen et al., 2019). Although progress has been made in identifying and validating specific miRNA and lncRNA targets in skeletal muscle cells, as well as elucidating the functional mechanisms of many miRNAs and lncRNAs in skeletal muscle, the correlations between miRNA, lncRNA, and the development of various muscles have not been fully explored. It is not understood how lncRNA interacts with miRNA to regulate skeletal muscle formation during sheep embryonic development, or which functional lncRNAs and miRNAs are differentially expressed during different embryonic stages.

To address these gaps in knowledge, this study examined the temporal expression profiles of sheep LD muscle lncRNAs and miRNAs at days 60, 90, and 120 of gestation, as well as at day 0 and 360 following birth. Using RNA sequencing, we were able to discover and add new lncRNAs to the sheep lncRNA and miRNA database. Our results will act as a resource for more thorough insight into the regulatory functions of lncRNAs in sheep, more detailed annotations of the sheep genome, and a better general understanding of mammalian skeletal muscle development.

Material and Methods

Ethics statement

All experimental protocols and procedures were approved by the Institutional Animal Care and Use Committee of Lanzhou Institute of Husbandry and Pharmaceutical Science of the Chinese Academy of Agricultural Sciences (approval no. NKMYD201805, dated 18 October 2018).

Animal and tissue samples

A total of 15 Gansu Alpine fine wool sheep were used in this study. All Gansu Alpine fine wool sheep were raised in the experimental facilities of Gansu Provincial Sheep Breeding Technology Extension Station (Huangcheng, Gansu, China) under the same conditions with free access to food and water in natural lighting. Caesarean sections were performed on three pregnant ewes from each developmental stage to collect female foetuses at 60, 90, and 120 days of gestation, and six female lambs were collected 0 and 360 days after birth. All animals were slaughtered after being anesthetized with xylazine chlorhydrate. We made all efforts to minimize animal suffering, and the slaughter procedures were carried out in accordance with animal welfare procedures. After slaughter, we collected three gestational (E60, E90, and E120) and two postnatal stage (D0 and D360) LD muscle samples. The tissue samples were snap frozen in liquid nitrogen and stored at −80 °C until analysis.

Muscle staining

We prepared the LD samples for histological sectioning using Carter & Clarke’s (1957) method. The LD samples were first placed in tubes containing 4% paraformaldehyde solution. After fixation, the samples were embedded, cut into slices, baked, H&E strained, and mounted (Auber, 1952). We examined the sections using a digital trinocular camera microscope (BA400Digital, McAudi Industrial Co., Ltd., Xiamen, China), and used the image analysis software Motic Images Advanced 3.2 to take and import an image. We then selected the objective lens magnification (40 ×) and the unit of measurement (um), used the polyline tool to measure the data, and exported the measured raw data in the .xls format for sorting and analysis.

RNA isolation, library construction, sequencing, and data analysis

The total RNA from 15 muscle samples was extracted using TRIzol reagent (Invitrogen, Carlsbad, CA, USA), according to the manufacturer’s instructions. We monitored RNA degradation and contamination using 1% agarose gels. RNA purity and concentration were detected using NanoDrop 2000 (Thermo Fisher Scientific Inc., Waltham, MA, USA), and were further measured using an Agilent 2100 bioanalyzer. Samples with an RNA integrity number (RIN) value greater than 8.0 were used for sequencing. RNA was digested by TruSeq stranded total RNA and a Ribo-Zero Gold Kit (Illumina, San Diego, CA, USA). After the total RNA was extracted, we removed the ribosome RNAs (rRNAs) to retain mRNAs and ncRNAs. The enriched mRNAs and ncRNAs were then broken down into short fragments using fragmentation buffer and turned into cDNA via reverse transcription and random primers. The second-strand cDNA was synthesized using DNA polymerase I, RNase H, dNTP (dUTP instead of dTTP), and buffer. Next, the cDNA fragments were purified using the Qiaquick PCR extraction kit, end-repaired, combined with poly A, and ligated to Illumina sequencing adapters. We used uracil-N-glycosylase (UNG) to digest second-strand cDNA that had been size-selected by agarose gel electrophoresis, amplified by PCR, and sequenced using an Illumina HiSeq TM 2500 from OE Biotechnology Corporation (Shanghai, China) (Zhou et al., 2016; Zhu et al., 2017). We pooled equal amounts of total RNA from muscle samples across different stages (i.e., E60, E90, E120, D0, and D360; n = 3) into one sample. To get high quality clean reads from the sequencing machines, we filtered out those that: (1) contained adapters, (2) had more than 10% unknown nucleotides (N), or (3) contained more than 50% low quality (Q-value ≤ 20) bases. We used the short reads alignment tool Bowtie 2 (Langmead & Salzberg, 2012) to map reads to the rRNA database, and then removed the rRNA mapped reads. The remaining reads were used in the transcriptome assembly and analysis. We then mapped the rRNA-removed reads from each sample to the reference genome using TopHat2 (Kim et al., 2013) and the following alignment parameters: (1) a maximum read mismatch of 2, (2) a 50 bp distance between mate-pair reads, and (3) a ± 80 bp error of distance between mate-pair reads. After aligning them with the reference genome, we re-aligned the unmapped reads (or very poorly mapped reads) using Bowtie2, and split the enriched unmapped reads into smaller groups to find potential splice sites. The section and section positions of these short segments were also predicted. We built a set of splice sites using initial unmapped reads from TopHat2 without relying on known gene annotation (Trapnell et al., 2010). The sequence alignments are not only useful for identifying expressed genes and their quantitative expressions, but also for identifying alternative splicing and new transcripts.

Transcript reconstruction was carried out using Cufflinks (Trapnell et al., 2012) which, together with TopHat2, allows biologists to identify new genes and new splice variants of known genes. We preferred to use the program’s reference annotation-based transcripts (RABT). Cufflinks constructed faux reads according to the reference gene to compensate for low coverage sequencing. During the last step of assembly, all reassembled fragments were aligned with reference genes and any similar fragments were removed. We used Cuff merge to combine transcripts from different replicas of a group into a comprehensive set of transcripts, and then to merge the transcripts from multiple groups into a comprehensive set of transcripts for further downstream differential expression analysis.

We quantified transcript abundances using RSEM (Li & Dewey, 2011). Transcript expression levels were normalized using the Fragments Per Kilobase of transcript per Million mapped reads (FPKM) method, and the differentially expressed transcripts of coding RNAs and lncRNAs were individually analysed. To identify differentially expressed transcripts across samples or groups, we used the edge R package (http://www.r-project.org/). We identified transcripts with a fold change ≥ 2 and a false discovery rate (FDR) <0.05 as significant DEGs. DEGs were then subjected to Gene Ontology (GO) function and KEGG pathway enrichment analysis. We performed gene expression pattern analysis to cluster genes of similar expression patterns from multiple samples (at least three from a specific time point, space, or treatment dose size order). To examine the DEG expression patterns, we normalized the expression data from each sample (in the order of treatment) to 0, log2(v1/v0), log2(v2/v0), and then clustered them using Short Time-series Expression Miner software (STEM) Ernst & BarJoseph, 2006. The clustered profiles with p-values ≤ 0.05 were considered significant, and the DEGs from each profile underwent GO and KEGG pathway enrichment analysis. Using the hypothesis test for p-value calculation and FDR correction (Saldanha, 2004), we defined GO terms or pathways with Q values ≤ 0.05 as significant enriched GO terms or pathways.

Real-time quantitative RT-PCR

The 15 muscle samples were stored at −80 °C prior to RNA extraction. We extracted the total RNA from the 15 muscle samples using TRIzol reagent (Invitrogen, Carlsbad, CA, USA). RNA purity and concentration were determined using a NanoDrop 2000 spectrophotometer (Thermo Fisher Scientific), and RNA integrity was evaluated using agarose gel electrophoresis staining with ethidium bromide.

Quantification was performed using a two-step reaction process: reverse transcription (RT) and PCR. Each RT reaction consisted of two steps. The first step was to combine 0.5 µg of RNA, 2 µl of 4 ×g DNA wiper Mix, and 8 µl of nuclease-free H2O. Reactions were performed in a GeneAmp® PCR System 9700 (Applied Biosystems, Foster City, CA, USA) for 2 min at 42 °C. The second step was to add 2µl of 5 × HiScript II Q RT SuperMix IIa. Reactions were performed in a GeneAmp® PCR System 9700 (Applied Biosystems) for 15 min at 50 °C, then for 5 s at 85 °C. The 10 µl RT reaction mix was then diluted ×10 in nuclease-free water and maintained at −20 °C. Real-time PCR was performed using a LightCycler® 480 II Real-time PCR Instrument (Roche, Basel, Switzerland) and 10 µl of a PCR reaction mixture that included 1 µl of cDNA, 5 µl of 2 ×ChamQ SYBR qPCR Master Mix, 0.2 µl of forward primer, 0.2 µl of reverse primer, and 3.6 µl of nuclease-free water. Reactions were incubated in a 384-well optical plate (Roche) at 95 °C for 30 s, followed by 40 cycles of 10 s at 95 °C, and 30 s at 60 °C. Each sample was run in triplicate for analysis. After the PCR cycles, we performed melting curve analysis to validate the specific generation of the expected PCR product. The primer sequences were designed and synthesized by Generay Biotech (Shanghai, China) using the lncRNA and miRNA sequences obtained from the NCBI database (Tables S15 and S16 ; (Bai et al., 2017). We estimated the PCR efficiency of each gene using standard curve calculation of the four cDNA serial dilution points. Cycle threshold (Ct) values were transformed to quantities using the comparative Ct method described by Chen et al. (2017). We carried out lncRNA and mRNA data normalization using the GAPDH reference gene, and miRNA data normalization using the U6 reference gene. The lncRNA and miRNA expression levels were normalized (using GAPDH and U6) and estimated using the 2−ΔΔCt method (Livak & Schmittgen, 2001).

Statistical analyses

After comparing the clean reads to the transcription template using Bowtie2 software (Langmead & Salzberg, 2012), we quantified the transcripts using eXpress software (Roberts & Pachter, 2013) to obtain FPKM values and mRNA and lncRNA counts. For samples without biological replicates, we calculated the p-value using the Audic_Claverie formula (Tiňo, 2009). miRNAs with p-values <  0.05 and TMP difference multiples >  2 were screened.

Results

Developmental changes in LD muscle across different stages

In this study, we stained the 12 LD in histological sections to analyze the area and circumference of muscle fibers at the various development stages from embryonic to postnatal. Our analysis showed that the area and circumferences of the muscle fibers did not increase over time, but it did decrease during the E90 and E120 periods (Fig. 1A). However, the tightness and evenness of muscle fibers significantly improved as the fetus continued to develop (Fig. 1B). These results suggest that embryonic skeletal muscle growth can be characterized by time.

Figure 1 HE stained longissimus dorsi tissue section.

(A) The trend chart of the muscle fibre area and circumference. (B) Comparison of the longissimus muscle of sheep foetuses at different developmental stages by H.E. staining. The 12 sheep LD were stained in histological sections to analyse the muscle fibre area and circumference at various stages. Statistics of measurements were analysed by one-way ANOVA with a Tukey’s test, * represents P < 0.05, ** represents P < 0.01.

LncRNA and small RNA sequencing

To comprehensively analyze sheep lncRNAs and miRNAs across different developmental stages, we constructed five cDNA libraries and five small RNA libraries (E60, E90, E120, D0, and D360) from three LD samples taken at 60, 90, and 120 days of gestation (E60, E90, and E120), and two from the postnatal (D0, D360) developmental stage. During lncRNA sequencing, we generated a total of 504,449,178 raw reads using all five libraries. After discarding adaptor sequences and low-quality reads, we obtained 477,031,266 clean reads (Table S1). Additionally, about 91.87% (91.70% - 92.48%) of the clean reads from each library were mapped to the sheep reference genome (Table S1), confirming the reliability of the sequencing data. During small RNA sequencing, we generated a total of 84,426,880 raw reads, and we obtained 10,623,459 - 26,521,572 clean reads (97%) ranging in size from 18 to 30 nt (Table S1). About 8,513,544 - 24,586,094 reads in the LW samples were perfectly mapped to the reference genome (GCF_000298735.2_Oar_v4.0), amounting to 72.2% - 92.7% of the clean reads (Table S2). Our sequencing results showed that most of the small RNA sequences were between 21 and 23 nt in length, which was consistent with the length distribution of the Dicer products and indicated that the sequencing results were good quality and could be used for follow-up analysis.

Differential expression analysis of sheep mRNAs and lncRNAs

To investigate the key mRNAs and lncRNAs involved in regulating sheep skeletal muscle development, we used RNA-seq datasets from five time points (three gestation stages and two postnatal stages) to characterize their time-specific expression patterns. When comparing the gene expression levels across the five developmental stages, we found 693 (352 upregulated) DEGs between E60 and E90 (Table S3), 799 (278 upregulated) DEGs between E90 and E120 (Table S4), 929 (672 upregulated) DEGs between E120 and D0 (Table S5), and 815 (257 upregulated) DEGs between D0 and D360 (Figs. 2A & 2B, Table S6). When analyzing these DEGs, we found that 478, 535, 633, and 580 genes were uniquely expressed in one of the two samples in E60 vs. E90, E90 vs. E120, E120 vs. D0, and D0 vs. D360, respectively (Fig. 2A). We detected two of these DEGs, FBN2 (gene ID: 101104991) and HR (gene ID: 443241), across all four comparisons. We also analyzed the differentially expressed lncRNAs (DE-lncRNAs) between E60 vs. E90, E90 vs. E120, E120 vs. D0, and D0 vs. D360, and detected 206, 239, 148, and 101 DE-lncRNAs, respectively (Fig. 2C & 2D, Tables S7–S10). We detected six of these DE-lncRNAs, TCONS_00009825, TCONS_00028121, TCONS_00122104, TCONS_00146167, TCONS_00189359, and TCONS_00195408, across all four comparisons. In the embryonic stage, the following gene expressions were significant: TCONS_00054028, TCONS_00023110, and TCONS_00150168 (Fig. 2E). Similarly, we found post-birth differential expression of other related genes, including TCONS_00016124, TCONS_00127579, TCONS_00009472, and TCONS_00014919 (Fig. 2F). Five differentially expressed lncRNAs were selected for qRT-PCR analysis (Table 1), and their differential expression was consistent with the RNA-Seq results. In summary, we found temporal-specific expression of the detected lincRNAs during the gestational and postnatal stages.

Figure 2 Number of DEG and DE-lncRNAs at different time stage comparisons.

(A) The number of DEGs across four comparisons: E60 vs. E90, E90 vs. E120, E120 vs. D0, and D0 vs. D360. (B) The number of up- and downregulated DEGs across four comparisons: E60 vs. E90, E90 vs. E120, E120 vs. D0, and D0 vs. D360. (C) The number of DE-lncRNAs across four comparisons: E60 vs. E90, E90 vs. E120, E120 vs. D0, and D0 vs. D360. (D) The number of up and downregulated DE-lncRNAs across the four comparisons: E60 vs. E90, E90 vs. E120, E120 vs. D0, and D0 vs. D360. The differentially expressed heatmap of significant lncRNA down (E) and upregulation (F).

Table 1 Validation of RNA-seq results by using quantitative RT-PCR.

QPCR indicates the expression level of the gene calculated by the 2−ΔΔCt method using quantitative RT-PCR, FPKM indicates the gene expression level calculated by sequencing.

Item	Genes	Method	E60	E90	E120	D0	D360	
miRNA	oar-miR-103	QPCR	1.79	2.35	1.86	1.00	1.29	
FPKM	21.13	612.71	53.16	3.39	17.04	
oar-miR-150	QPCR	0.01	0.01	0.03	1.00	1.03	
FPKM	1.27	3.73	13.77	3.39	17.04	
oar-miR-362	QPCR	0.80	1.52	1.43	1.00	0.12	
FPKM	1376.64	2959.89	2391.22	879.08	354.87	
oar-miR-410-3p	QPCR	1.14	1.90	1.97	1.00	0.00	
FPKM	68.99	175.89	198.56	32.31	0.09	
oar-miR-221	QPCR	1.49	1.29	1.65	1.00	1.67	
FPKM	46.88	52.56	80.03	34.52	91.31	
LncRNA	TCONS_00144061	QPCR	28.44	28.18	28.73	1.00	0.88	
FPKM	22416.88	21299.49	25556.43	15182.36	10449.53	
TCONS_00105394	QPCR	0.57	0.64	0.84	1.00	1.09	
FPKM	489.56	561.35	950.09	1454.33	2306.80	
TCONS_00105227	QPCR	159.79	337.79	205.07	1.00	0.11	
FPKM	2137.33	2720.73	2231.06	423.75	66.20	
TCONS_00091985	QPCR	0.26	0.49	0.80	1.00	1.18	
FPKM	546.08	557.27	590.00	1311.02	1469.90	
TCONS_00091984	QPCR	0.05	0.04	0.03	1.00	0.72	
FPKM	148.81	230.36	202.90	987.67	572.94	

GO and KEGG pathway analysis

We used KEGG pathway analysis of DE-lncRNA target genes to identify the pathways that were enriched in DE-lncRNA target genes. When comparing E60 vs. E90, E90 vs. E120, E120 vs. D0, and D0 vs. D360, we found that in the most significantly enriched pathways, DE-lncRNA target genes participated in signal transduction, the endocrine system, the nervous system, cell growth and death folding, sorting and degradation, the immune system, cellular community eukaryotes, translation, amino acid metabolism, and the carbohydrate metabolism pathways (Fig. 3). The top 20 significantly enriched KEGG analyses for each comparison’s DE-lncRNAs are shown in Fig. S1. GO enrichment analysis of DE-lncRNA targeted genes also delivered a large number of significant annotations, but in order to examine temporal changes in skeletal muscle development, we provided the detailed results from four adjacent DE-lncRNA comparisons. In the E60 vs. E90 comparison, T=the top 30 GO terms that were significantly related to genes targeted by total lncRNAs included male meiotic nuclear division, mitotic spindle assembly checkpoint, kinetochore, cell, protein kinase activity, and microtubule plus −end binding (Fig. S2). In the E90 vs. E120 comparison, the most significantly enriched GO terms for the total lncRNAs were associated with muscle development, and included signal peptide processing, activation of MAPKK activity, L −type voltage −gated calcium channel complex, voltage −gated calcium channel complex, histone acetyltransferase complex, receptor signaling complex scaffold activity, and Rho GTPase binding (Fig. S3). In the E120 vs. D0 comparison, the top 30 processes for down and upregulated lncRNAs were associated with male meiotic nuclear division, cellular response to leukemia inhibitory factor, intercellular bridge, and integral component of endoplasmic reticulum membrane (Fig. S4). In the D0 vs. D360 comparison, we found associations with male meiotic nuclear division, cellular response to leukemia inhibitory factor, microtubule organizing center, and integral component of endoplasmic reticulum membrane (Fig. S5). Across the four comparisons, we observed significant changes in skeletal muscle development during the prenatal and neonatal stages.

Figure 3 The top KEGG enrichment analyses of the differentially expressed lncRNAs in E60 vs. E90 (A), E90 vs. E120 (B), E120 vs. D0 (C), and D0 vs. D360 comparisons (D).

LncRNA-gene interaction network construction

We extracted the candidate sequences by screening out lncRNAs and genes that were not on the same chromosome as the candidate targets. We used the RNA interaction software RIsearch (Wenzel, Akbaşli & Gorodkin, 2012) to predict the binding of candidate lncRNAs and genes at the nucleic acid level, with the number of bases directly interacting with each other according to the two nucleic acid molecules was no less than 10, and the free energy of base binding was no more than -50. The possible regulatory interaction networks between lncRNAs and their target genes (mRNAs) were constructed. In the E60 vs. E90 comparison, the lncRNA-gene interaction network was comprised of 15 lncRNAs and 66 protein-coding genes with close networks; among these, TCONS_00196403 and TCONS_00196407 constructed complex network relationships with targeted coding genes (Fig. 4A). In the E90 vs. E120 comparison, the lncRNA-gene interaction network was made up of complex network nodes and lncRNA-gene connections between 34 lncRNAs and 79 protein-coding genes. TCONS_00029967 and TCONS_00088616 had a strong correlation with protein-coding genes (Fig. 4B). In the E120 vs. D0 comparison, the lncRNA-gene interaction network was comprised of complex network nodes and lncRNA-gene connections between 25 lncRNAs and 95 protein-coding genes. TCONS_00181895, TCONS_00149972, and TCONS_00098734 had a strong correlation with protein-coding genes (Fig. 4C). In the D0 vs. D360 comparison, the lncRNA-gene interaction network was comprised of complex network nodes and lncRNA-gene connections between 11 lncRNAs and 86 protein-coding genes. TCONS_00196411, TCONS_00157638, and TCONS_00126396 had a strong correlation with protein-coding genes (Fig. 4D).

Figure 4 The lncRNA-gene network for the comparisons of E60 vs. E90 (A), E90 vs. E120 (B), E120 vs. D0 (C), and D0 vs. D360 (D).

Temporal-specific expression of the detected sheep miRNAs

The miRNA expression level comparisons across the five developmental stages revealed that there were 328 (265 upregulated) DEGs between E60 and E90 (Table S11), 302 (243 upregulated) DEGs between E90 and E120 (Table S12), 295 (166 upregulated) DEGs between E120 and D0 (Table S13), and 184 (110 upregulated) DEGs between D0 and D360 (Fig. 5A, Table S14). We found that the miRNA expression levels during the embryonic and postnatal stages differed significantly. During the embryonic stage, the differentially expressed miRNAs were miR-134, miR-2387, miR-105, miR-3957, miR-493, miR-541, miR-767, miR-432, and miR-433 (Fig. 5B). We also conducted a differential analysis of the miRNAs based on their expression levels. The miRNAs with expression levels that changed more than two times and with P values <0.05 were deleted. The results showed that some miRNAs were differentially expressed after birth, specifically miR-22, miR-365, miR-3556, miR-29, miR-193, miR-150, miR-1, and miR-133 (Fig. 5C). Five differentially expressed miRNAs were selected for qRT-PCR analysis (Table 1), and their differential expression was consistent with the RNA-Seq results. We generated a heatmap based on the expression patterns at all time points and across all known expressed miRNAs (Fig. 5D). The significantly enriched processes and observed pathways were consistent with the temporal changes in skeletal muscle development across all sampled stages.

Figure 5 Sheep LD muscle miRNA features.

(A) The number of up- and downregulated miRNAs across the four comparisons: E60 vs. E90, E90 vs. E120, E120 vs. D0, and D0 vs. D360. The differentially expressed heatmap of significant miRNA down (B) and upregulation (C). (D) A heatmap of miRNAs across five-time stages. Each row represents the expression levels of all detected miRNAs.

miRNA target gene prediction and functional analysis

Most animal miRNAs were not completely complementary to their target mRNA, mainly in the 3′ non-coding region (3′ UTR) of the target mRNA, and their mechanism of action is translation inhibition. Previous studies also found that animal miRNAs could target the 5′ end of the mRNA as well as the coding region (Aleksandra et al., 2013). In this study, we used the miRanda algorithm (John et al., 2004) to predict miRNA target genes. We detected 216, 143, 198, and 204 differentially expressed miRNAs and 33,106, 26,431, 29,049 and 27,554 target genes in the E60 vs. E90, E90 vs. E120, E120 vs. D0, and D0 vs. D360 comparisons, respectively. The high expression of novel 129_mature in E60 vs. E90 indicates that the expression product of the target AHNAK2 gene is related to AHNAK nucleoprotein 2 and transcript variant X1, and is involved in the nucleus, cytoplasm, cytosol, plasma membrane, Z disc, T-tubule, cytoplasmic vesicle membrane, and sarcolemma pathways. The high expression of novel 410_mature in E90 vs. E120 suggests that the expression product of the target AHNAK2 gene is mainly related to the S-phase response (cyclin-related). The high expression of novel 360_mature in E120 vs. D0 and D0 vs. D360 shows that the expression product of the target PASD1 gene is related to the PAS domain containing 1 and transcript variant X1, and is involved in the transcription coactivator binding, nucleus, nuclear speck, negative regulation of circadian rhythm, negative regulation of transcription, DNA-templated, rhythmic process, and Cry-Per complex pathways. We performed GO enrichment analysis at GO Level 2 for all target genes and the genes of differentially expressed miRNAs. The biological processes involved included cellular process, biological regulation, metabolic process, response to stimulus, and regulation of biological process (Figs. S6 & S7).

Discussion

In this study, we systematically described the lncRNA and miRNA succession processes during the three prenatal stages and two postnatal stages in sheep embryo LD. We found that lncRNA and miRNA expression were time-specific, and that the differentially expressed miRNAs at the embryonic stage and after birth were also different. miRNAs played a regulatory role during multiple stages of skeletal muscle development, and participated in skeletal muscle stem cell proliferation, differentiation, migration, and resting, myoblast proliferation and differentiation, muscle fiber type conversion, energy metabolism, and other processes (Sun et al., 2010). For example, we found high expression abundance of miR-1 after birth. Previous studies found that the miR-1′s biological function is to promote myogenic differentiation. The main target genes were Hdac4, Pax3, Pax7, Notch3, Hdac2, ND1, and Cox1 (Sun et al., 2010). In C2C12 cells, miR-1 complements the mitochondrial oxidative phosphorylation-related genes Cox1 and ND1 when promoting target gene translation and upregulating mitochondrial energy metabolism (Zhang et al., 2014). Additionally, we found high expression abundance of miR-432 during the embryo stage. Previous studies found that miR-432 inhibits the PI3K/AKT/mTOR signaling pathway by targeting E2f3 and P55pik genes, while simultaneously inhibiting myoblast proliferation and differentiation (Ma et al., 2017). These results indicate that different miRNAs play important roles in myoblast proliferation and differentiation at different embryonic stages.

In ruminants, the prenatal stage is crucial for skeletal muscle development because almost all muscle fibers are formed during this period, not after birth (Du et al., 2010). Previous livestock studies have shown that many miRNAs are highly expressed during the prenatal stages and possibly during skeletal myogenesis (Li et al., 2011; Qin et al., 2013). miRNAs regulate myogenic differentiation by directly inhibiting myogenic transcription factors. For example, miR-186 inhibits terminal muscle differentiation by targeting myogenesis, and miR-181 enhances MyoD activity by inhibiting the negative regulator of the myogenic differentiation antigen Hox A11 (Antoniou et al., 2014). Currently, the main myo-miRs are miR-1, miR-133, and miR-206. miR-1 and miR-133 are expressed in both myocardium and skeletal muscle, while miR-206 is expressed only in skeletal muscle (Eva et al., 2007; Sempere et al., 2004). The main functions of miR-206 are to inhibit myoblast proliferation and promote its differentiation. The main functions of miR-133 are to promote myoblast proliferation, inhibit differentiation (Wang, 2013), and contribute to cell fate determination and muscle regeneration. miR-208, miR-499, and miR-486 are also classified as muscle-specific miRNAs. miR-133 targets Runx2, Trps1, and Prdm16, which are responsible for osteoblast, chondrocyte, and fat cell development, respectively (Ying et al., 2012).

We observed significant upregulation of miR-133 when comparing D0 vs. D360 and E120 vs. D0, indicating that miR-133 inhibits cell differentiation in other directions, and is thus conducive to skeletal muscle development (Hang et al., 2013). When comparing E120 vs. D0 and E60 vs. E90, we found significant upregulation of the highly-expressed miRNAs miR-127 and miR-136. miR-127 is mainly expressed in skeletal muscle and has been found to be upregulated during C2C12 and satellite cell (SC) differentiation (Zhai et al., 2017). Therefore, it is conceivable that miR-127 is involved in muscle development and functional SC postnatal muscle regeneration processes. miR-136 promotes vascular muscle cell proliferation through the erk1/2 pathway by targeting ppp2r2a in atherosclerosis (Zhang et al., 2015). It has also been suggested that by playing a role in regulating vascular myocyte proliferation during early life, miR-136 ensures the normal differentiation and growth of muscle fibers.

In this study, we found that miR-22, miR-365, miR-3556, miR-29, miR-193, miR-150, miR-1, and miR-133 were differentially expressed during the embryonic stage, and that miR-22 expression was highest in skeletal muscle and gradually upregulated during mouse myoblast cell differentiation. miR-22 overexpression repressed C2C12 myoblast proliferation and promoted myoblast differentiation into myotubes, whereas miR-22 inhibition showed the opposite results (Wang et al., 2018). miR-22 expression in the longissimus tissues of adult pigs was found to be higher than in 33 and 65 day-old pig embryos (Huang et al., 2008), and our study also found higher miR-22 expression in adult sheep than in sheep embryos. Additionally, miR-365 is located on chromosome 16p13.12 and is involved in cell proliferation and apoptosis in many types of cells (Nie et al., 2012). A previous study found that miR-365 significantly inhibits myoblast cell activity and cell growth, suggesting that miR-365 can markedly suppress duck myoblast proliferation (Sun et al., 2019). In this study, we found that miR-365 expression was higher in adult sheep than in sheep embryos, indicating that the gene may be involved in inhibiting the myoblast proliferation. We also observed that oar-miR-410-3p expression was very high during the embryonic stage, which was in agreement with another study that found that miR-410-3p overexpression suppressed invasion, migration, and proliferation, downregulated EMT-associated molecule expression, and promoted apoptosis and apoptotic factor expression in rhabdomyosarcoma cells (Zhang et al., 2019). There have been many functional studies on the four lncRNAs, linc-MD1 (Legnini et al., 2014; Yoon, Abdelmohsen & Gorospe, 2013), Yam-1 (Li et al., 2018; Simionescu-Bankston & Kumar, 2016, sirt1AS (Ming et al., 2016; Wang et al., 2016), and H19 (Dey, Pfeifer & Dutta, 2014; Gao et al., 2014), involved in muscle development. These lncRNAs affect myogenic differentiation by interacting with one or more miRNAs. Linc-MD1 is a muscle-specific lncRNA that acts as a competitive RNA in mouse and human myoblasts and as a sponge of miR-133 and miR-135, which regulate Maml1 and Mef2c expression. linc-MD1 downregulation inhibits and linc-MD1 overexpression promotes muscle differentiation. H19 is abundantly expressed in embryonic tissues, is inhibited after birth, and is only expressed in skeletal muscle. H19 exon 1 encodes miR-675, which is an miRNA that induces myogenic differentiation expression. In this study, we found that the downregulation of involved miRNAs regulated the progress of muscle development and biological adhesion, and activated myoblasts during differentiation and fusion between day 60 and 90 of gestation. This result was consistent with the muscle cell generation stages found in fetal sheep and goats (Guo et al., 2016; Picard et al., 2002). Intramuscular fat adipogenesis occurs during the late gestational stage in pigs and sheep, and using the GO processes, we found that target genes were enriched between E120 and D0. This suggests that in goats, miRNA is involved in the formation of intramuscular fat during the prenatal stage.

In order to provide new insights into lncRNA and miRNA function during sheep skeletal development, we systematically described the lncRNAs and miRNAs that regulate sheep skeletal myogenesis and examined their LD muscle temporal expression profiles from gestation to the yearling stage. However, our method involved pooling three muscle samples from each stage into one sample to sequence, which had certain flaws. Before selecting this method, we searched the relevant pool-seq literature in detail, and found that: (1) genome sequencing pools of individuals is a cost-effective approach to determining genome-wide allele frequencies in an unbiased manner from a large number of individuals; (2) once the minimum quality criteria have been met, pool -seq-based allele frequency estimates are accurate and reliable; and (3) pool-seq has been successfully applied across a wide range of bulk segregant analyses, evolution and resequencing studies, evolutionary genome analyses, and time-series data and cancer genomics analyses (Schlötterer et al., 2014). Additionally, we performed real-time quantitative RT-PCR verification of the differential genes obtained by sequencing, which validated the time-specific lncRNA and miRNA expression patterns and the accuracy of the gene expression quantification.

Conclusion

In this study, we systematically identified the lncRNAs and miRNAs that regulate sheep skeletal myogenesis, and examined their LD temporal expression profiles from the gestational to postnatal stages. We described a set of lncRNAs, miRNAs, and genes related to LD muscle growth across five developmental stages. Additionally, we provided four visual lncRNA-gene regulatory networks that can be used to further explore lncRNA function in sheep. We also described several lncRNAs that interact with miRNAs to regulate myogenic differentiation. Our results are valuable resources for future studies on lncRNA and miRNA biology, particularly those regarding sheep muscle, and are helpful in understanding lncRNA and miRNA function in sheep. Integrating published data on lncRNAs and miRNAs and their influence on skeletal muscle development in goats and sheep is a critical step towards building a database.

Supplemental Information

Supplemental Information 1 LncRNA sequence statistics for the 5 samples

Click here for additional data file.

Supplemental Information 2 Small RNA sequence statistics for the 5 samples

Click here for additional data file.

Supplemental Information 3 Differentially expressed mRNAs in E60 and E90 comparisons

Click here for additional data file.

Supplemental Information 4 Differentially expressed mRNAs in E90 and E120 comparisons

Click here for additional data file.

Supplemental Information 5 Differentially expressed mRNAs in E120 and D0 comparisons

Click here for additional data file.

Supplemental Information 6 Differentially expressed mRNAs in D0 and D360 comparisons

Click here for additional data file.

Supplemental Information 7 Differentially expressed lncRNAs and their corresponding target genes in E60 and E90 comparisons

Click here for additional data file.

Supplemental Information 8 Differentially expressed lncRNAs and their corresponding target genes in E90 and E120 comparisons

Click here for additional data file.

Supplemental Information 9 Differentially expressed lncRNAs and their corresponding target genes in E120 and D0 comparisons

Click here for additional data file.

Supplemental Information 10 Differentially expressed lncRNAs and their corresponding target genes in D0 and D360 comparisons

Click here for additional data file.

Supplemental Information 11 Differentially expressed miRNAs and their sequence in E60 and E90 comparisons

Click here for additional data file.

Supplemental Information 12 Differentially expressed miRNAs and their sequence in E90 and E120 comparisons

Click here for additional data file.

Supplemental Information 13 Differentially expressed miRNAs and their sequence in E120 and D0 comparisons

Click here for additional data file.

Supplemental Information 14 Differentially expressed miRNAs and their sequence in D0 and D360 comparisons

Click here for additional data file.

Supplemental Information 15 LncRNA Primers for qRT-PCR

Click here for additional data file.

Supplemental Information 16 miRNA Primers for qRT-PCR

Click here for additional data file.

Supplemental Information 17 The top 20 KEGG enrichment analyses of total differentially expressed lncRNAs in E60 and E90 (A), E90 and E120 (B), E120 and D0 (C), and D0 and D360 (D) comparisons

Click here for additional data file.

Supplemental Information 18 GO annotation for predicted target genes of differentially expressed lncRNAs in E60 and E90 comparisons

Click here for additional data file.

Supplemental Information 19 GO annotation for predicted target genes of differentially expressed lncRNAs in E90 and E120 comparisons

Click here for additional data file.

Supplemental Information 20 GO annotation for predicted target genes of differentially expressed lncRNAs in E120 and D0 comparisons

Click here for additional data file.

Supplemental Information 21 GO annotation for predicted target genes of differentially expressed lncRNAs in D0 and D360 comparisons

Click here for additional data file.

Supplemental Information 22 Comparison of miRNA target genes and all genes at GO Level2 level distribution of E60 vs. E90 and E90 vs. E120

Click here for additional data file.

Supplemental Information 23 Comparison of miRNA target genes and all genes at GO Level2 level distribution of E120 vs. D0 and D0 vs. D360

Click here for additional data file.

We would like to thank Fanwen Li, Jigang Liu, Yazhou Wen, and Tianzhao Luo from the Gansu Provincial Sheep Breeding Technology Extension Station for their assistance in sample collection.

Additional Information and Declarations

Competing Interests

Author Contributions

Animal Ethics

DNA Deposition

Data Availability

The authors declare there are no competing interests.

Chao Yuan conceived and designed the experiments, performed the experiments, analyzed the data, prepared figures and/or tables, authored or reviewed drafts of the paper, and approved the final draft.

Ke Zhang and Xiaolong Wang analyzed the data, prepared figures and/or tables, authored or reviewed drafts of the paper, and approved the final draft.

Yaojing Yue, Tingting Guo, Jianbin Liu, Chune Niu, Xiaoping Sun and Ruilin Feng performed the experiments, authored or reviewed drafts of the paper, and approved the final draft.

Bohui Yang conceived and designed the experiments, authored or reviewed drafts of the paper, and approved the final draft.

The following information was supplied relating to ethical approvals (i.e., approving body and any reference numbers):

All experimental protocols and procedures were approved by the Institutional Animal Care and Use Committee of Lanzhou Institute of Husbandry and Pharmaceutical Science of Chinese Academy of Agricultural Sciences (Approval No. NKMYD201805; dated: 18 October 2018).

The following information was supplied regarding the deposition of DNA sequences:

The data is available at NCBI Sequence Read Archive (SRA): SRP188484.

The following information was supplied regarding data availability:

Data are available at NCBI GEO: SRR8731864–SRR8731873.

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
