# Peer review of "Analysis of dynamic and widespread lncRNA and miRNA expression in fetal sheep skeletal muscle"

_PeerJ, doi:10.7717/peerj.9957_

## Round 0.1 · original submission · Major Revisions

Thank you for submitting your manuscript to PeerJ. Based on the comments of three reviewers I invite you to resubmit after making major revisions. Most significantly, please address the concerns of all three reviewers that no biological replicates were included and that identified genes were not validated. Reviewers also requested additional detail in your methods section. Note that reviewer 2 included some of their suggestions as an attached document.

As you review your methods section please note that PeerJ requires that qPCR studies follow MIQE guidelines (https://www.ncbi.nlm.nih.gov/pubmed/19246619). The legend for your Table 1 could also use more information about the values in the table (units?, definition?, etc.)

Please be sure to respond to all reviewer comments in a response letter with your resubmission.

·

Basic reporting

The English language needs to be improved.

Experimental design

The methods section should be clear and detailed enough. For instance, methods of GO analysis, KEGG analysis, interaction network construction, and statistical analysis of qRT-PCR results, etc. More details are needed in this section.

Validity of the findings

no comment

Additional comments

Line 26: sheep is an important……

Line 32: …longissimus dorsi (LD)…, italic longissimus dorsi in manuscript.

Line 43: …targeted by the different miRNAs…

Line 71: …TncRNA???, not clear.

Line 125: …detected using…, and digested by…

Lines 123, 125, 126, 136, 137, etc: Company information should be consistently written.

Line 129-130: “Equal amounts of total RNA from the muscle samples at the different stages (i.e. E60, E90, E120, D0, and D360; n=3) were pooled into one sample”? Is it means one sample for each stage? Why ?? No biological duplication, is that reasonable?

Line 151-152: How to measure and analyze the area and circumference of muscle fibers?? Provide more essential details.

Line 194, line 254: “The expression of them was consistent with the RNA-Seq results”, Table 1 only provides the qRT-PCR results. How to analyze the qRT-PCR expression and sequencing results consistent?

Line 222: Using the RNA interaction software RIsearch…., Software requires references.

Line 330: “…the 4 lncRNAs, linc-MD1, Yam-1, sirt1AS and H19”, provide references.

The discussion could be further strengthen, highlight the findings of this study, and a clear significance of the study given.

You state in the data availability that the dataset were available on NCBI under BioProject PRJNA526287. Usually, BioProject ID does not indicate that RNA-Seq data were uploaded successfully, the data needs to be submitted to the SRA (Sequence Read Archive) database to obtain the corresponding accession number (starting with SRR or SRA) after obtaining the BioProject and BioSample ID. Please provide the SRA accession number in this section.

Reviewer 2 ·

Basic reporting

The study is interesting but there are main flaws about the method (pooled samples without biological replicates).

Experimental design

The main flaws were “samples at the different stages were pooled into one sample” and “without biological replicates”.

Validity of the findings

No validation experiment about any target gene or ncRNA.

Annotated reviews are not available for download in order to protect the identity of reviewers who chose to remain anonymous.

Reviewer 3 ·

Basic reporting

The Materials and Methods were too ambiguous and too simple.

Experimental design

No replicates were used for RNA-seq and miRNA-seq

Validity of the findings

The DE lncRNAs, miRNA and mRNAs should be validated

Additional comments

Yuan et.al, performed RNA-seq and miRNA-seq to perform the lncRNA and miRNA dynamic profiling during skeletal muscle development at 5 stages. Differentially expressed lncRNAs and miRNAs were identified, and regulatory networks were constructed. This work might be useful for understanding sheep muscle development. However, various critical issues should be addressed.
1. Your most important issue is that apparently no experiment was performed using replicates, making it hard to assess if any of the observed differences are consistent and reproducible. If not possible, at least, some DEGs should be validate by RT-PCT?
2. The next most important item is that the description of Materials and Methods section was too simple. For examples, what types of sequencing library were constructed? Whether it’s strand-specific or not? how the lncRNAs and miRNAs were identified and analyzed? How the authors identified differentally expressed lncRNAs and miRNAs? What criterions were used? What software the authors were used to predict the targets miRNAs? What the sex of the prenatal sheep? So many important information were missed in this section, make me very difficult to understand the results.
3. How many samples were used for analyzing the tightness and evenness of muscle fibers in Figure 1B?
4. What is long miRNAs in the legend of Figure 5

---

## Round 0.2 · Minor Revisions

Thank you for submitting your revised manuscript to PeerJ and for your responses to the three reviewers. Based on the comments of reviewer 3 and my own reading of your paper I invite you to resubmit after making some minor revisions.

Your methods section for the qPCR procedure still needs additional information to approach MIQE guidelines. Please include the following details: How were tissues treated/stored prior to RNA purification?; what kit/technique was used to synthesize cDNA and how much RNA was used?; what RNA equivalent was used in each qPCR reaction?; what were the cycling conditions and reaction conditions for the qPCR?; what enzyme and detection system was used?; were technical replicates used?; what range in Cq values were considered acceptable in technical triplicates?; were primers tested for efficiency?; what software was used to do your expression analysis?. Similar to reviewer 3’s comments, please explain in the discussion section how the lack of biological replicates affects the interpretation of your results.

Your new supplemental tables (S15 and S16) with the qPCR primer sequences are formatted in two different ways, and S16 has a heading for product length, but the length is not given. Please add that information and I suggest standardizing the format of these two tables. If you have efficiency percentages calculated for each primer pair these tables would be a good place to list that information.

Please edit the new text in the Muscle Staining methods section to use the past tense. In the RNA isolation section you mention that RNA integrity was measured with a NanoDrop, but I believe that equipment can only measure purity and concentration. Please also check that citations are properly inserted by your citation management software.

The legend for your Table 1 could still more clearly state what these values indicate (gene expression levels?) and explain the column headings.

Lastly, it looks like one more author has been added to the paper since the first submission, but no new experiments appear to have been done. Can you please indicate the new contribution made by this author?

Please be sure to respond to the comments above and those from reviewer 3 in a response letter with your resubmission. I note that in response to reviewer 3’s question about why samples were pooled instead of analyzing them as biological replicates you cited a paper by Schlötterer et al. to support this approach. I would suggest explaining that within your discussion and citing this paper for support. Please also explain whether that reasoning extends to the qPCR experiments.

·

Basic reporting

no comment

Experimental design

no comment

Validity of the findings

no comment

Reviewer 2 ·

Basic reporting

no comment

Experimental design

no comment

Validity of the findings

no comment

Reviewer 3 ·

Basic reporting

The quality of the revised manuscript has been largely improved. However, several questions still need to be improved.
1. The legends of Figures should provide more information. For instance, in fig.1B, the numbers of pigs and the statistical method used for this analysis should be added. In fig.3, we can't see P value of each KEGG pathway.
2. In table 1, the FPKM or TPM values of each genes should also be listed for better comparison.
3. The written English of the manuscript needs improvement. For examples, in conclusion section, both present and past tenses are used. The tenses need to be unified。

Experimental design

There are no biological replicates for the RNA-seq dataset. If the authors could't add new samples, at least, the authors should discuss this shortages of the experimental design in the Discussion section.

Validity of the findings

no comment

---

## Round 0.3 · Minor Revisions

Thank you for submitting a revision of your paper. Based on comments by reviewer 3 and my own reading of the resubmission I invite you to submit a new version with minor revisions.

All reviewers are now satisfied with your experimental design. I do have some suggestions that I think would enhance the readability of your paper and more strongly convey your key findings to your readers.

The discussion includes a great deal of introductory information, with your key findings not mentioned until the third page. I would consider reviewing your introduction and discussion to determine if there is redundant information, or if some of the material in the discussion can be moved to the introduction to better justify and explain your research question. The discussion could be stronger if you led with the key findings of your paper.

Thank you for including an explanation and justification for pooling your samples. However, I suggest that you move this new section later in the discussion after you have explained your key findings.

I believe that the current wording of the introduction makes it unclear which of the studies you describe are past studies, and how they lead to your current study. Clarifying this could help to explain the rationale and gap in knowledge being filled by this current work.

You mention in your revised methods that efficiencies were calculated for qPCR primers. Please include these efficiency percentages in the supplemental tables for the primers or somewhere else in your manuscript

Could you please explain in your next rebuttal letter why the new author was added? Was this due to new work added in revision or where they accidentally left off the initial paper?

Lastly, I believe that your paper still needs editing for style. For example, there are a number of run-on sentences that need to be addressed throughout the paper. PeerJ has an editing service if that would be useful.

Reviewer 3 ·

Basic reporting

The revised manuscript have greatly improved. I agree to accept this manuscript.

Experimental design

no comment

Validity of the findings

no comment

---

## Round 0.4 · Minor Revisions

Thank you for addressing my last set of comments and for submitting your revised manuscript. I believe that your manuscript is now ready for publication pending final edits to the text. You mentioned an interest in using PeerJ’s editing services, and the editorial office will be in contact with you about those services. You are of course also welcome to use any other editing services you wish to make grammatical and stylistic improvements to the manuscript.

As you make these edits I suggest that you divide the second paragraph of the discussion into multiple paragraphs. Please also correct how the micro RNAs are noted, as sometimes you use “mir”, other times “miR”, and names include both dashes and underscores. Please correct these to be consistent and in line with the notation in the literature.

---

## Round 0.5 · accepted · Accept

Thank you for the edits and the resubmission of your manuscript. I am happy to now accept your paper for publication in PeerJ.

You will be given the option to make the reviews of your manuscript available to readers. Please consider doing so as this review record can be a great resource for readers of your paper and contributes to more transparent science.

Thank you for choosing PeerJ as a venue for publishing your work.